# Do alexithymia and negative affect predict poor sleep quality? The moderating role of interoceptive sensibility

Yun-Hsin Huang[1], Chien-Ming Yang[2,3]*, Ya-Chuan Huang[2], Yu-Ting Huang[3], Nai-Shing Yen[2,3]*

1 Department of Psychology, Fo Guang University, Jiaoxi Township, Yilan, Taiwan, 2 Department of Psychology, National Chengchi University, Taipei, Taiwan, 3 Research Center for Mind, Brain, and Learning, National Chengchi University, Taipei, Taiwan

* yangcm@nccu.edu.tw (CMY); nsy@nccu.edu.tw (NSY)

**Data Availability Statement:** Data cannot be shared publicly because it is the part of the "Human Project from Mind, Brain and Learning Database." Data are available from the Research

## Abstract

### Objectives

Emotion-related hyperarousal is an important core pathology of poor sleep. Studies investigating the interplay of alexithymia and affective experiences in determining sleep quality have yielded mixed results. To disentangle the inconsistency, this study examined the concurrent predictive power of alexithymia, and negative and positive affect, while incorporating interoceptive sensibility (IS) as a possible moderator.

### Methods

A sample of 224 (70.10% were female) participants completed the Toronto Alexithymia Scale, Positive and Negative Affect Schedule, Pittsburgh Sleep Quality Index, Multidimensional Assessment of Interoceptive Awareness (MAIA), and Marlowe-Crowne Social Desirability Scale (for controlling response bias) using paper and pencil. A two-stage cluster analysis of the MAIA was used to capture IS characteristics. Stepwise regression was conducted separately for each IS cluster.

### Results

A three-group structure for IS characteristics was found. Higher alexithymia was predictive of poor sleep quality in the low IS group, while higher negative affect predicted poor sleep quality in the moderate and high IS groups. Additionally, alexithymia and positive affect were significantly different in the three IS groups, while negative affect and sleep quality were not.

### Conclusions

Emotion and cognitive arousal may impact sleep quality differently in individuals with different levels of internal focusing ability, depending on physiological versus emotional self-conceptualization. The implications on pathological research, clinical intervention, study limitations and future directions are discussed.

Center for Mind, Brain, and Learning of National Chengchi University (contact via https://rcmbl.nccu.edu.tw/index.php?option=com_content&view=article&id=10&Itemid=106&lang=en-us or rcmbl@nccu.edu.tw) for researchers who meet the criteria for access to confidential data.

**Funding:** This work was supported by the Ministry of Education of Taiwan (Award Number 108H-121-02), which was not involved in the actual study. The funders had no role in study design, data collection, and analysis, publication decision, or manuscript preparation.

**Competing interests:** The authors have declared that no competing interests exist.

# Introduction

## Emotion and sleep quality

Hyperarousal is an important pathological mechanism in sleep disturbances [1–4], emotion plays an important role in psychopathology, and negative affect has been shown to be associated with poor sleep [5, 6] and insomnia [7]. Compared with good sleepers, poor sleepers experience more negative affect and arousal at night and more negative affect during the day [8]. Cellini et al. [9] suggested that one's emotional state is tightly associated with a high level of arousal, which mainly reduces the quality of sleep.

In contrast, being unaware of one's emotions may also be associated with the mechanisms of hyperarousal and sleep quality. A large study hypothesized that insomnia patients tend to internalize or inhibit negative emotions, which in turn leads to physiological arousal that causes sleep problems [10]. Clinically, many insomniac patients do not recognize their sleep problems as "emotional" in nature, but rather a "physiological" symptom. Unawareness of emotions differs experientially from a strong negative affect, but both are related to hyperarousal and sleep quality. The interplay between such contradictory but relevant phenomena has not yet been well established.

## Alexithymia and sleep quality

To investigate the relationship between unawareness of emotions and sleep quality, the concept of *alexithymia* may be probed. Specifically, alexithymia refers to the difficulty in identifying and describing emotional feelings as well as the tendency to not incorporate emotions in relevant cognitive processes [11, 12]. It has been hypothesized to be associated with higher tonic physiological arousal due to persistent difficulty in emotion regulation [13, 14]. Some evidence has shown higher levels of arousal among high-alexithymia individuals under neutral or baseline conditions [15–17] and in the recovery phase [18]. This persistent tonic hyperarousal may further affect sleep quality.

Empirical evidence supports an association between alexithymia and sleep disturbances related to hyperarousal. Alexithymia has been consistently found to be associated with insomnia [19, 20], insomnia symptoms [21, 22], and poor sleep quality [23]. Hyyppä et al. [24] further found that alexithymia was simultaneously associated with poorer sleep and hypersecretion of cortisol, which supports the hypothesis that a tendency to suppress (or repress) psychological conflicts may lead to increased persistent tonic physiological arousal and further sleep disturbances.

However, when other emotional constructs are considered, study results have been inconsistent. In some studies, after controlling for self-reported depressive [25] and anxiety [26] scores, the relationship between alexithymia and sleep quality disappeared. In contrast, evidence has also indicated that the association between alexithymia and sleep disturbances persists after controlling for or simultaneously incorporating depression [27, 28] or one's general mood [21]. Taken together, these findings suggest that the association between alexithymia and sleep disturbances cannot be completely attributed to other emotional problems, although some interplay exists.

Two possible considerations may be examined to address the mixed results mentioned above. First, alexithymia is conceptualized as "there being emotion unrecognized/unexpressed" when it is used to infer psychopathology mechanisms [12, 24, 29]. However, both recognized and unrecognized emotions may play a role in the development of depression [30] and anxiety [31, 32]. Examining the concurrent effects of alexithymia and depression may not be the best way to examine the emotional pathway to sleep disturbances due to this overlap.

Therefore, investigating the concurrent effects of alexithymia and daily affective experiences (in purely emotional terms) may be helpful. Second, individual differences may also play a role. Honkalampi and Saarinen [33] found that sleep disturbances were associated with alexithymia in men but with depression in women, suggesting that there might be differential functions of concurrent alexithymia and depression among different populations. There may be an underlying mechanism that moderates the effects of alexithymia on other psychological processes, and how an individual is aware, perceives, and manipulates bodily sensations could be a candidate, as this is highly relevant to the interplay of the physical and psychological aspects of emotion.

### Interoceptive sensibility

Interoceptive sensibility (IS) is helpful to differentiate the effects of negative affect and alexithymia. It refers to the tendency to be internally focused as well as the belief and manipulation of the sensed internal state [34–36]. A cross-cultural study reported that somatization was significantly associated with alexithymia among participants from high-alexithymic cultures (i.e., Asian Americans and Malaysians) but not among participants from low-alexithymic cultures (i.e., European Americans) [37]. Researchers generally agree that among Asian cultures, especially among the Chinese, people tend to identify and communicate distress somatically rather than emotionally [38, 39]. Thus, LeBerenbaum and Raghavan [37] suggested that alexithymia is more strongly associated with affect problems in Asian cultures than in European or American cultures. In other words, when emotional stimuli cause physiological reactions but are not processed as emotional, these reactions can further lead to problems related to arousal via "alexithymia." Instead, when these reactions are processed as emotional, such reactions may lead to arousal-related problems via "negative emotions." Thus, it is possible that IS plays a moderating role in hyperarousal-related processes.

### Current study

This study aimed to investigate the hyperarousal-related psychopathological mechanisms in sleep disturbances associated with alexithymia and negative affectivity in individuals with different IS characteristics. We hypothesized that, in individuals with a tendency to be internally focused and process bodily sensations emotionally (i.e., higher IS), physiological arousal may interfere with sleep quality in the expression of negative affect. In contrast, in individuals with a tendency to ignore or interpret bodily sensations non-emotionally (i.e., lower IS), physiological arousal may interfere with sleep quality in the expression of alexithymia. Thus, we used self-reported questionnaires to assess alexithymia, daily affective experiences, IS, and sleep quality among our community sample. Stepwise regression analysis of alexithymia, negative and positive affectivity on sleep quality in individuals with different IS characteristics was performed. We hypothesized that (1) a higher negative affect predicts worse sleep quality in individuals with higher IS tendency and (2) higher alexithymia predicts worse sleep quality in individuals with lower IS tendency.

## Materials and methods

### Participants

The data are part of a larger project, the Human Project from Mind, Brain, and Learning (HPMBL) in Taiwan. HPMBL concerns comprehensive well-being of adults in urban life, and was approved by the Research Ethics Committee of Chengchi University (NCCU-REC-201810-I074). Online advertisements were posted on a university discussion webpage on social

network media. The inclusion criteria for HPMBL were: 1) age 20 to 64; 2) native speakers of traditional Chinese; 3) normal vision (with or without correction); and 4) not diagnosed as having mental disorders that influence reality testing or cognitive ability, such as schizophrenia or dementia. A total of 249 participants participate the HPMBL; 25 participants had missing questionnaire data for this study. Hence, a final sample of 224 participants was included in this analysis. The mean age of final sample was 22.13 years (SD = 2.71), and there were 157 (70.10%) women and 67 (29.90%) men.

## Measurements

**Pittsburgh Sleep Quality Index (PSQI).**    The PSQI is a 19-item self-rated questionnaire which was used to assesses sleep quality and disturbances over a 1-month time period [40]. The PSQI contains seven component scores: subjective sleep quality, sleep latency, sleep duration, habitual sleep efficiency, sleep disturbances, use of sleeping medication, and daytime dysfunction. The sum of component scores generates the global score, which ranges from 0 to 21; higher score indicates poorer sleep quality. The traditional Chinese version of the PSQI (CPSQI) had an overall reliability coefficient of 0.82–0.83, and acceptable test-retest reliability (coefficient 0.85) [41].

**Toronto Alexithymia Scale (TAS).**    This widely used tool for alexithymia was developed by Bagby, Parker [11]. This 20-item questionnaire is scored on a 5-point Likert scale; total scores range from 20 to 100, with higher scores representing a stronger tendency to alexithymia. The TAS has three subscales: difficulty identifying feelings (DIF, e.g., I do not know what's going on inside me), difficulty describing feelings (DDF, e.g., I am able to describe my feelings easily), and externally oriented thinking (EOT, e.g., I prefer talking to people about their daily activities rather than their feelings). The TAS has good internal consistency, test-retest reliability, adequate convergent and concurrent validity, and a stable factor structure [42–44]. The traditional Chinese version of TAS had a good internal consistency and construct validity comparable to the original version [45].

**Positive and Negative Affect Schedule (PANAS).**    The widely used PANAS assesses daily experience of positive (PA) and negative affect (NA) [46]. This 20-item questionnaire is scored on a 5-point Likert scale, with a higher summed score of each subscale indicating higher PA or NA in daily experience for a selected duration ("recent couple weeks" in this study). PANAS has been found to have good psychometric properties among various samples [46, 47]. The Cronbach's alpha of each subscale of the traditional Chinese version was ranged from 0.76 to 0.94 across various samples in Taiwan [48, 49].

**Multidimensional Assessment of Interoceptive Awareness (MAIA).**    MAIA is a self-report questionnaire for IS that has been translated into various languages [36]. The 32-item MAIA is rated on a 6-point Likert scale (0 to 5). It contains 8 scales, including: (1) noticing: awareness of uncomfortable, comfortable, and neutral body sensations (e.g., I notice when I am uncomfortable in my body); (2) not-distracting: tendency not to ignore or distract oneself from sensations of pain or discomfort (e.g., I distract myself from sensations of discomfort); (3) not-worrying: tendency not to worry or experience emotional distress with sensations of pain or discomfort (e.g., I can notice an unpleasant body sensation without worrying about it); (4) attention regulation: ability to sustain and control attention to body sensations (e.g., I can return awareness to my body if I am distracted); (5) emotional awareness: awareness of the connection between body sensations and emotional states (e.g., I notice how my body changes when I am angry); (6) self-regulation: ability to regulate distress by attending to body sensations (When I feel overwhelmed, I can find a calm place inside); (7) body listening: active listening to the body for insight (e.g., When I am upset, I take time to explore how my body

feels); (8) trusting: experience of one's body as safe and trustworthy (e.g., I am at home in my body). A higher mean score on each scale indicates a stronger tendency to IS in that domain. The internal consistency of each scale in the original version ranged from 0.66 to 0.82 [36]. The traditional Chinese version has moderate to good internal consistency (Cronbach's alpha ranged from 0.46 to 0.91) and construct validity comparable to the original version [50].

**Marlowe-Crowne Social Desirability Scale-short form.** This 13-item forced-choice scale is the short form of the widely used Marlowe-Crowne Social Desirability Scale [51]. A higher score indicates a greater tendency to a biasing response for social approval. The short form has good psychometric properties and is highly correlated with its original version [51, 52].

## Procedure

The database project HPMBL was commenced in 2018. Multilevel constructs were collected using self-reported questionnaires and computerized tests. All the data collection was offline, such that questionnaires were all responded with paper and pencil. The first wave of data collection included two rounds of computerized tests and questionnaire filling, each requiring one to two hours. Data collected in Wave two were reduced; thus, one round each of computerized tests and questionnaire filling were executed. After obtaining written informed consent, the first round of data collection was conducted and then appointed for the next data collection. In Wave one, participants received 400 NTD (about 13 USD) for each first-round computerized tests and questionnaires, and 600 NTD (about 20 USD) for the second round. In Wave two, they received 500 NTD (approximately 17 USD) for each round of data collection.

## Statistical analysis

Partial correlations between all measures after controlling for social desirability were performed. Two-stage clustering analysis of eight subscale scores using agglomerative hierarchical clustering (Ward's method) was performed in stage 1, and K-means (K = 3 according to the stage 1 result) in stage 2. Differences in the TAS, PANAS, and PSQI among the three IS groups were compared using ANCOVA and post hoc contrasts. Hierarchical regression using social desirability was entered in step 1, and TAS, PA, NA was entered using stepwise regression in step 2. The statistical analyses were conducted using SPSS Windows software version 21.

## Results

### Characteristics of measures and three IS clusters

The overall Cronbach's alpha of PSQI was 0.89; 0.74 for TAS; 0.84 for PANAS-PA and 0.90 for PANAS-NA; 0.91 for MAIA; 0.68 for Marlowe-Crowne Social Desirability Scale. According to the final centroids (Table 1), the three IS clusters were categorized as low, moderate, and high IS groups. There was no significant difference of demographic data between the three groups. TAS and PANAS-PA scores showed statistically significant difference, such that the high IS group exhibited lower TAS and higher PANAS-PA; PANAS-NA and PSQI were not significantly different among the groups (Table 2). We also tested the gender differences on all measures controlling for social desirability. Males are higher on the MAIA-Attention Regulation ($F = 4.05$, $p = .045$), MAIA-Self-Regulation ($F = 4.39$, $p = .037$), and TAS-EOT ($F = 6.37$, $p = .012$). All other measures were not significantly different from male to female.

After controlling for social desirability, except for the two reversed scales (N-D and N-W), the MAIA subscales were profoundly related to each other. TAS was associate with PANAS-PA and PANAS-NA; in contrast, PANAS-PA and PANAS-NA was not associated. PSQI was significantly associated with TAS, PANAS-PA, and PANAS-NA (Table 3).

**Table 1. Final cluster centroids of three IS clusters.**

|  | Low IS | Moderate IS | High IS | F | p |
|---|---|---|---|---|---|
| **N** | 3.03 | 3.55 | 3.84 | 30.96 | < .001 |
| **N-D** | 2.14 | 1.86 | 2.00 | 3.05 | .049 |
| **N-W** | 1.63 | 1.45 | 1.80 | 4.33 | .014 |
| **AR** | 2.01 | 2.69 | 3.48 | 136.32 | < .001 |
| **EA** | 2.56 | 3.29 | 4.10 | 97.44 | < .001 |
| **S-R** | 1.72 | 2.68 | 3.58 | 150.16 | < .001 |
| **BL** | 1.69 | 2.79 | 3.89 | 194.79 | < .001 |
| **T** | 2.41 | 3.63 | 4.06 | 88.01 | < .001 |

$^{*}p < .05$

$^{***}p < .001$

**N**, MAIA Noticing; **N-D**, MAIA Not-Distracting; **N-W**, MAIA Not-Worrying; **AR**, MAIA Attention Regulation; **EA**, MAIA Emotion Awareness; **S-R**, MAIA Self-Regulation; **BL**, MAIA Body Listening; **T**, MAIA Trusting

## Regression analyses: Constructs predicting PSQI among three IS clusters

The regression coefficients of the three IS clusters are given in Table 4. After controlling for social desirability, higher TAS scores were significantly predictive of higher PSQI ($\beta = 0.36$, $p = .010$) in the low IS cluster, while PA and NA scores were selected out. In contrast, higher NA scores were significantly predictive of higher PSQI in the moderate ($\beta = 0.26$, $p = .011$) and high ($\beta = 0.35$, $p = .004$) IS clusters, while TAS and PA scores were selected out.

**Table 2. Sample characteristics and differences of measures between three IS clusters.**

|  | All participants N = 224 | Low IS (L) N = 59 | Moderate IS (M) N = 92 | High IS (H) N = 73 |  |  |
|---|---|---|---|---|---|---|
|  | N (%) | N (%) | N (%) | N (%) | $\chi^2$ |  |
| **Male** | 67 (29.90%) | 12 | 30 | 25 | 3.55 |  |
| **Female** | 157 (70.10%) | 47 | 62 | 48 |  |  |
|  | M ± SD | M ± SD | M ± SD | M ± SD | F | Contrasts |
| **Age** | 22.13 ± 2.71 | 22.07 ± 1.78 | 22.32 ± 2.35 | 22.13 ± 2.71 | 0.28 |  |
| **Education year** | 15.43 ± 1.55 | 15.39 ± 1.47 | 15.72 ± 1.73 | 15.43 ± 1.55 | 2.36 |  |
|  | M ± SD | M ± SD | M ± SD | M ± SD | $F^a$ | Contrasts |
| **TAS** | 49.91 ± 10.64 | 52.93 ± 11.00 | 51.82 ± 8.74 | 45.05 ± 10.98 | 7.91*** | L ≈ M > H[b] |
| **TAS-DIF** | 18.13 ± 5.43 | 19.03 ± 5.81 | 18.80 ± 4.63 | 16.56 ± 5.78 | 2.19 |  |
| **TAS-DDF** | 13.75 ± 3.80 | 14.42 ± 4.01 | 14.35 ± 2.38 | 12.45 ± 3.96 | 4.51* | L ≈ M > H |
| **TAS-EOT** | 18.02 ± 3.84 | 19.47 ± 3.56 | 18.66 ± 3.41 | 16.04 ± 3.80 | 12.22*** | L ≈ M > H |
| **PANAS-PA** | 26.67 ± 6.50 | 23.69 ± 5.85 | 25.75 ± 5.72 | 30.25 ± 6.38 | 15.98*** | L ≈ M < H |
| **PANAS-NA** | 22.52 ± 7.71 | 24.46 ± 8.53 | 21.73 ± 6.35 | 21.96 ± 8.37 | 1.71 |  |
| **PSQI** | 5.98 ± 2.72 | 6.20 ± 3.09 | 6.12 ± 2.50 | 5.63 ± 2.67 | 0.28 |  |

$^{*}p < .05$

$^{***}p < .001$

**TAS**, Toronto Alexithymia Scale; **PANAS-PA**, Positive and Negative Affect Schedule–Positive Affect; **PANAS-NA**, Positive and Negative Affect Schedule–Negative Affect; **PSQI**, Pittsburgh Sleep Quality Index

[a] ANCOVA between three IS clusters with controlling for social desirability

[b] L, Low IS group; M, Moderate IS group; H, High IS group

**Table 3. Partial correlation of modelled variables (controlling for social desirability).**

| | N | N-D | N-W | AR | EA | S-R | BL | T | TAS | PANAS-PA | PANAS-NA |
|---|---|---|---|---|---|---|---|---|---|---|---|
| **N-D** | 0.01 | | | | | | | | | | |
| **N-W** | -0.11 | 0.10 | | | | | | | | | |
| **AR** | 0.37*** | -0.04 | 0.23*** | | | | | | | | |
| **EA** | 0.52*** | -0.01 | -0.12 | 0.61*** | | | | | | | |
| **S-R** | 0.28*** | -0.10 | 0.13 | 0.64*** | 0.49*** | | | | | | |
| **BL** | 0.34*** | -0.06 | -0.10 | 0.58*** | 0.64*** | 0.66*** | | | | | |
| **T** | 0.23*** | -0.09 | -0.03 | 0.45*** | 0.36*** | 0.47*** | 0.48*** | | | | |
| **TAS** | -0.12 | -0.14 | -0.14* | -0.18** | -0.13 | -0.22** | -0.14* | -0.27*** | | | |
| **PANAS-PA** | 0.29*** | -0.05 | -0.09 | 0.32*** | 0.39*** | 0.34*** | 0.29*** | 0.26*** | -0.33*** | | |
| **PANAS-NA** | 0.02 | -0.03 | -0.26*** | -0.15* | 0.06 | -0.13 | -0.01 | -0.16* | 0.17* | -0.12 | |
| **PSQI** | 0.09 | 0.01 | -0.16* | -0.13 | 0.05 | -0.10 | 0.03 | -0.22** | 0.18** | -0.21** | 0.25*** |

*$p < .05$

**$p < .01$

***$p < .001$

**N**, MAIA Noticing; **N-D**, MAIA Not-Distracting; **N-W**, MAIA Not-Worrying; **AR**, MAIA Attention Regulation; **EA**, MAIA Emotion Awareness; **S-R**, MAIA Self-Regulation; **BL**, MAIA Body Listening; **T**, MAIA Trusting; **TAS**, Toronto Alexithymia Scale; **PANAS-PA**, Positive and Negative Affect Schedule–Positive Affect; **PANAS-NA**, Positive and Negative Affect Schedule–Negative Affect; **PSQI**, Pittsburgh Sleep Quality Index

## Discussion

Our findings support the hypothesis that bodily sensations may affect sleep quality via alexithymia and negative affectivity depending on the nature of internal focus. In the low IS group, higher alexithymia significantly predicted worse sleep quality, while higher negative affectivity significantly predicted worse sleep quality among the moderate and high IS groups. To our knowledge, this is the first study to investigate the moderating effect of IS on the emotion-related psychopathological mechanisms of sleep disturbances and their theoretical and clinical implications.

In our study, group differences in alexithymia and positive affect provided some insight into emotional processing based on previous theoretical considerations. Alexithymia levels

**Table 4. Regression coefficients among three IS clusters.**

| Low IS | | | | | Moderate IS | | | | | High IS | | | | |
|---|---|---|---|---|---|---|---|---|---|---|---|---|---|---|
| **Predictor** | **B** | **β** | **t** | **p** | **Predictor** | **B** | **β** | **t** | **p** | **Predictor** | **B** | **β** | **t** | **p** |
| **Step 1** | | | | | **Step 1** | | | | | **Step 1** | | | | |
| SDS | -0.15 | -0.13 | -1.00 | .322 | SDS | -0.07 | -0.06 | -0.60 | .551 | SDS | -0.31 | -0.29 | -2.59 | .012 |
| $R^2 = .02$; $Adj. R^2 = .00$ $F(1, 57) = 1.00$; $p = .322$ | | | | | $R^2 = .00$; $Adj. R^2 = -.01$ $F(1, 90) = 0.36$; $p = .551$ | | | | | $R^2 = .09$; $Adj. R^2 = .07$ $F(1, 71) = 6.71$; $p = .012$ | | | | |
| **Step 2a** | | | | | **Step 2a** | | | | | **Step 2a** | | | | |
| SDS | -0.00 | -0.00 | -0.01 | .995 | SDS | -0.06 | -0.05 | -0.50 | .621 | SDS | -0.17 | -0.16 | -1.37 | .176 |
| TAS | 0.10 | 0.36 | 2.67 | .010 | TAS | - | - | - | - | TAS | - | - | - | - |
| PANAS-PA | - | - | - | - | PANAS-PA | - | - | - | - | PANAS-PA | - | - | - | - |
| PANAS-NA | - | - | - | - | PANAS-NA | 0.10 | 0.26 | 2.58 | .011 | PANAS-NA | 0.11 | 0.35 | 2.97 | .004 |
| $R^2 = .13$; $Adj. R^2 = .10$; $\Delta R^2 = .11$ $F(2, 56) = 4.11$; $p = .022$ | | | | | $R^2 = .07$; $Adj. R^2 = .05$; $\Delta R^2 = .07$ $F(2, 89) = 3.52$; $p = .034$ | | | | | $R^2 = .19$; $Adj. R^2 = .17$; $\Delta R^2 = .10$ $F(2, 70) = 8.15$; $p = .001$ | | | | |

**SDS**, Marlowe-Crowne Social Desirability Scale-Short Form; **TAS**, Toronto Alexithymia Scale; **PANAS-PA**, Positive and Negative Affect Schedule–Positive Affect; **PANAS-NA**, Positive and Negative Affect Schedule–Negative Affect

were significantly higher in the low- and moderate-IS groups than in the high-IS group (Table 2). This is consistent with previous findings that alexithymia is associated with an interceptive awareness deficit [35, 53, 54], supporting the view that alexithymia is a marker of atypical interoceptive awareness [55]. Interestingly, PA significantly differed among the IS groups, while NA did not, which indicates that internal focusing ability may influence the level of positive but not negative affectivity in daily life. It is possible that positive experiences are more of a simple "feeling" than negative experiences, in which cognitive processes are evidently highly involved [56–58]. Further, TAS was more strongly correlated with PA than NA (Table 3). Given the high emphasis on emotion regulation in the case of alexithymia, the difficulty in recognizing negative emotions in alexithymia is strongly embedded in its conceptualization, although this has not been directly referred to in previous literature [13]. However, relatively little is known about positive emotions, although an association between alexithymia and positive affect-related deficiencies has been found [59, 60]. In light of our findings on the differences between IS groups and the relationship between PA, NA, and TAS, the difficulty in recognizing positive emotions in alexithymia is worthy of more attention.

The differential predictive power of alexithymia and negative affectivity may provide a possible explanation for the previous inconsistency and theoretical relationship between alexithymia and sleep disturbances. Among individuals who tended not to notice or use bodily sensations in an internally oriented manner, alexithymia lead to problematic sleep (Table 4). This finding is in line with evidence that alexithymia is predictive of sleep disturbances after controlling for affective constructs [21, 27, 28]. In contrast, in individuals who were able to process such sensations in a more internally focused way and use these in emotional self-regulation, sleep quality was influenced by affect rather than by alexithymia (Table 4). Given that TAS was positively correlated with PSQI in our overall sample (Table 2), this finding is more consistent with the view that the predictive power of alexithymia in the case of sleep disturbances is attenuated after controlling for affective construct [25, 26]. Taken together, the moderating effect of IS may explain this inconsistency.

These implications raise additional clinical concerns. Among individuals with low IS, a tendency to alexithymia may lead to the conceptualization of "physiological symptoms" as sensations of somatic arousal; hence, such sleep problems may be experienced as "physiological" [17]. External-oriented thinking may result in cognitive arousal dissociated from somatic feelings, such that thinking content is mainly related to stressors; that is, generalized worry about how things are might constitute the prevalent form of cognitive arousal [61] among individuals with low IS. In contrast, among individuals with higher IS, somatic hyperarousal may be experienced in an effective way, such that the individual is aware that somatic arousal is emotion-related and therefore feels a strong negative emotion [5–7]. As with the ability to be internally focused, depressive rumination focusing on emotional distress [62] might be a major component of cognitive arousal. To address repetitive thinking and somatic arousal, insomnia interventions have suggested improving one's emotion regulation ability [63, 64]. However, alexithymia tendency and negative affectivity should be addressed differently according to the patient's internal focusing ability (i.e., IS). For those who can process these sensations in a more emotional manner, typical cognitive behavioral therapy for insomnia may be the first choice for enhancing sleep quality [65, 66]. However, for those who tend not to process bodily sensations internally, interventions addressing alexithymia that enhance emotion regulation through expressive writing [67], mindfulness [68], or mentalization-based techniques [69] should be incorporated. Mindfulness has already been effectively incorporated in insomnia interventions [70]; future studies should investigate patient-intervention fitness according to IS.

There are several strengths and limitations of this study. By incorporating IS, this study raised fresh insights on psychopathology and interventions for sleep disturbances. The

individual difference of IS provide a possibility to explain the inconsistent findings of alexithymia, negative affect and sleep problem. Accordingly, individualized intervention for sleep problems can be designed. Apart from the main focus of the study, we have also found a potential differentiation between PA and NA in their association strength with somatic feelings. The participants were community samples recruited through online advertisements. Hence, self-selection bias was unavoidable. A higher homogeneity of participants may also lead to other problems in addition to a lack of generalizability. We did not find significant gender differences in self-reported negative affectivity, alexithymia, or sleep quality, as in previous studies [71–73]. This may be due to the high homogeneity that overrides gender differences. In sum, this study's ecological validity is not ideal. Although the regression results in our study were statistically significant, the $R^2$s were all small, indicating a small effect of the studied variables. Sleep is a complex process influenced by the circadian, homeostatic, and arousal processes [2, 74, 75]. Emotion and alexithymia are only a part of the arousal process; hence, the contribution might in fact be small.

Several directions can be applied in future studies. First, IS is a subcomponent of interoception, the "sensing the physiological condition of the body, as well as the representation of the internal state within the context of ongoing activities, and is closely associated with motivated action to homeostatically regulate the internal state" [76, p.693]. Other domains of interoception can be investigated. Similarly, there are multiple levels and measures of affect and sleep quality. Incorporating different levels of such construct, especially physiological assessments of interoception (e.g., brain imaging of insula, heart-evoked potential heartbeat detection task), affect (e.g., brain and autonomic reactions of emotion induction), and sleep (e.g., polysomnography) can provide important information. Second, emotion regulation is an important factor involved with arousal-related sleep problem intervention. Additionally, emotion regulation is embedded in the construct of alexithymia [13, 14], intimately related to negative affective experience [32], and highly relevant to interoceptive awareness [36]. Future study may incorporate emotion regulation in addition to the constructs of this study. Third, sleep and interoception may interplay within aspects other than hyperarousal. For example, interoception is also directly linked to homeostasis [76], another determined process of sleep quality [75]. Some possibilities of the relations between sleep and interoception have been discussed, however more empirical studies are required to determine such interplay [77]. Beyond the relationship between sleep and interoception, our finding further suggests that individual differences of interoception is associated with the express of differential mechanism of sleep process. This phenomenon can be examined in homeostatic relevant aspects of sleep.

## Conclusion

We tested a novel idea that how an individual process their own bodily sensations associates with sleep disturbances using self-reported questionnaires. Alexithymia predicts lower sleep quality among low IS group. In contrast, negative affectivity predicts lower sleep quality among moderate and high IS groups. Further examination of psychopathology and intervention of sleep disturbance incorporating IS should be implemented in future.

## Author Contributions

**Conceptualization:** Yun-Hsin Huang, Chien-Ming Yang.

**Data curation:** Yun-Hsin Huang, Yu-Ting Huang.

**Formal analysis:** Yun-Hsin Huang.

**Funding acquisition:** Nai-Shing Yen.

**Methodology:** Yun-Hsin Huang.

**Project administration:** Yun-Hsin Huang, Yu-Ting Huang, Nai-Shing Yen.

**Supervision:** Chien-Ming Yang, Nai-Shing Yen.

**Writing – original draft:** Yun-Hsin Huang, Chien-Ming Yang, Ya-Chuan Huang.

**Writing – review & editing:** Yun-Hsin Huang, Chien-Ming Yang, Nai-Shing Yen.

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
