## [Decision Letter · Decision Letter 0]

15 Jun 2022

PONE-D-21-13329

Do Alexithymia and Negative Affect Predict Poor Sleep Quality? The Moderating Role of Interoceptive Sensibility

PLOS ONE

Dear Dr. Yen,

Thank you for submitting your manuscript to PLOS ONE. After careful consideration, we feel that it has merit but does not fully meet PLOS ONE’s publication criteria as it currently stands. Therefore, we invite you to submit a revised version of the manuscript that addresses the points raised during the review process.

Please find the reviewer reports from two reviewers below. Both reviewers have asked for additional clarifications in the methodology and presentation of the study.

We look forward to receiving your revised manuscript.

Kind regards,

Hanna Landenmark

Staff Editor

PLOS ONE

Journal Requirements:

3. Thank you for stating the following in the Acknowledgments Section of your manuscript: "This work was supported by "The Human Project from Mind, Brain and Learning" of NCCU from the Higher Education Sprout Project by the Ministry of Education in Taiwan."

Please remove any funding-related text from the manuscript and let us know how you would like to update your Funding Statement. Currently, your Funding Statement reads as follows: "This work was supported by Ministry of Education of Taiwan, and was not involved with any industry. The funders had no role in study design, data collection and analysis, decision to publish, or preparation of the manuscript."

Reviewers' comments:

Reviewer's Responses to Questions

**Comments to the Author**

1. Is the manuscript technically sound, and do the data support the conclusions?

Reviewer #1: Partly

Reviewer #2: Yes

2. Has the statistical analysis been performed appropriately and rigorously? 

Reviewer #1: No

Reviewer #2: Yes

3. Have the authors made all data underlying the findings in their manuscript fully available?

Reviewer #1: Yes

Reviewer #2: Yes

4. Is the manuscript presented in an intelligible fashion and written in standard English?

Reviewer #1: No

Reviewer #2: Yes

5. Review Comments to the Author

Reviewer #1: ID: PONE-D-21-13329

Title: Do Alexithymia and Negative Affect Predict Poor Sleep Quality? The Moderating Role of Interoceptive Sensibility

Thank you for providing a chance to review this manuscript.

Comment:

Major revision.

Detailed information:

Introduction

Your introduction takes up nearly half of the entire article. Please write more concisely and distill your key points.

Page 10-11 Line172-181: The purpose of the research in the introduction is very similar to the abstract. Please describe the corresponding theoretical support and hypotheses, and highlight the practical significance of your research.

Page 10-11 Line176, “IS (MAIA)”: This is really confusing.

Materials and Methods

Participants

Page 11 Line186-187, "3) normal vision (with or without correction)": Is this inclusion criterion necessary? And how do you manage the missing data?

Measurements

Page 11: You mentioned PSQI is a Chinese version, it’s a traditional one or simplified one? And how about the other measures? Did you use Chinese version too? If so, have they been validated? What’s the measurement properties of them? They should be concisely reported with details.

Procedure

Page 14: What is the response rate for obtaining data through online surveys? How do you control data quality?

Results

Page 20 Table 4: What is the meaning of R² if it’s too small? Furthermore, what is meaning of the regression model considering all R²s are small?

Discussion

Page 21-22 Line 321-347: The “discussion” should be resented with the order of “results” are presented. What’s more, this part is a bit confusing, please be more organized and refined.

Taking all the comments above into consideration, this paper is interesting and written with a lot of work, but some issues may still need to be reconsidered. I hope you can further adjust the construct of your content. And it is of great importance to request a native English speaker to check your writing and make the expression more understandable.

My bests.

Your reviewer.

Reviewer #2: The current study submitted to PLOS One aimed to investigate hyperarousal-related psychopathological mechanisms in sleep disturbances associated with alexithymia and negative affectivity, as well as the possible moderating role of interoceptive sensibility in a population-based sample. The study found that interoceptive sensibility tendency moderates the psychopathological mechanism of poor sleep quality. Further, the study concluded that the group differences in alexithymia and positive affect provided further insight into emotional processing based on previous theoretical considerations.

In general, the study gives a thorough scientific background on the subject and uses rigid subjective methods with many standardized scales (PSQI; TAS; MAIA; PANAS and more) to explore the relationship.

This study adds a small contribution to the existing literature on sleep and different emotional and behavior traits; however, a few points should be addressed before publication.

Abstract

The number of females participants is around 70% and instead of stating that in numbers in the methods section (abstract), it would be beneficial to the reader to see the percentage.

Introduction

The introduction gives a thorough review covering previous work on topics such as hyperarousal, emotions, alexithymia, interoceptive sensibility, and how they intertwine with sleep quality, insomnia, and overall sleep disturbances.

It struck me that just the introduction is roughly 2000 words (8 pages) and 80 references. Scientific papers usually tend to aim for 500–1000 words in the introduction and I highly recommend shortening the introduction (possibly move to methods, discussion, and cut down on the word count). Further, I would recommend limiting the excess number of references.

Materials and Methods

It would be of use to state where participants were recruited from? Was the online advertisement in a large city? University setting? Urban or rural area?

It would be beneficial to report what software the statistical analysis were run in and add the information to the Statistical Analysis section.

Results

In my opinion the lack of participants characteristic table is a problem. The authors jump straight into Partial Correlation of Measures (Table 1) but there is missing information regarding participants characteristics, score on the scales and possible some background information on the participants (age, height, social status) if that was collected in the study.

Further, a short description on the gender difference and how they performed on the test would also give the reader a better insight into the results. For example: did females report more alexithymia? Where there any gender differences regarding Pittsburgh Sleep Quality Index scores?

Discussion

The reviewer would like a short sentence regarding the possible self-selection bias which is always a risk when studies use general population samples chosen via advertisement.

Further a discussion is needed regardingthe gender split of the participants. Females where 70% participants and previous research has shown that there is gender difference in both sleep behavior and mental aspect such as reporting of anxiety, alexithymia and depression.

Other comments

Line 63 – instead of saying “some studies” it can be written as other studies in order to prevent repetitiveness (from line 61 and line 65).

Line 343 – Recommend using effective instead of affective in the sentence: “somatic hyperarousal may be experienced as an affective way”

Line 371 – there is NA ., and than a large letter after the , that should to be fixed.

6. PLOS authors have the option to publish the peer review history of their article (what does this mean?). If published, this will include your full peer review and any attached files.

Reviewer #1: No

Reviewer #2: No

---

## [Author Response · Author response to Decision Letter 0]

7 Jul 2022

The responses to the editor were presented in the Cover Letter. The responses to the reviewers were presented in the Response to Reviewers file due to its length. We thank all the comments from the editor and reviewers.

---

## [Decision Letter · Decision Letter 1]

16 Aug 2022

PONE-D-21-13329R1Do Alexithymia and Negative Affect Predict Poor Sleep Quality? The Moderating Role of Interoceptive SensibilityPLOS ONE

Dear Dr. Yen,

Thank you for submitting your manuscript to PLOS ONE. After careful consideration, we feel that it has merit but does not fully meet PLOS ONE’s publication criteria as it currently stands. Therefore, we invite you to submit a revised version of the manuscript that addresses the points raised during the review process.

We look forward to receiving your revised manuscript.

Kind regards,

Runtang Meng, Ph.D., M.M., M.B.B.S.

Guest Editor

PLOS ONE

Journal Requirements:

Additional Editor Comments:

Dear Dr. Nai-Shing Yen,

I have received the external reviews of the manuscript ("Do Alexithymia and Negative Affect Predict Poor Sleep Quality? The Moderating Role of Interoceptive Sensibility") you submitted to PLoS One.

You will find the comments and suggestions of the reviewers below.

Based on the advice received and my own review, I have decided that your manuscript can be reconsidered for publication if you are prepared to incorporate minor revisions.

I ask that in preparing your revised manuscript you consider all comments carefully. Please check online for eventual reviewer attachments.

In submitting your revised manuscript, please include a cover letter giving specific details as to how you addressed each comment along with the page numbers where changes appear.

Reviewer 1

ID: PONE-D-21-13329

Title: Do Alexithymia and Negative Affect Predict Poor Sleep Quality? The Moderating Role of Interoceptive Sensibility.

I appreciate your efforts to improve the manuscript and to respond to the comments made in the first review process. However, there are still minor issues that need to be addressed.

Comment: minor revision.

Detailed information:

Abstract

This section is still too long, especially the "Objectives" part, please simplify it appropriately.

Materials and methods

Participants

Line 167, page 7, “The university is in a large city”: This means nothing to me and the readers. I suggest removing it.

Line 173 – 174, page 8: All number digits should be unified. Check other places too.

This section is not very fluent and seems like several different parts spliced together. I would suggest combining and trimming some of the sentences to make them easier to read.

Measurements

Line 185-186, page 8; line 199, page 9; line 239, page 11: The reliability of this study should be placed in the "Results" section.

Procedure

Line 250-253, page 11: "NTD" or "NTDs"? Please express it uniformly.

Results

Line 267 – 269, page 12: Please state vital results in the tables and change these two sentences with "(Table 1)" and "(Table 2)", which may make the expression clearer.

Line 270 – 271, page 12: Please standardize the number of decimals appearing in the whole text.

Table 1 – Table 4: 1) All scientifical tables should be formatted as three-line tables. 2) All abbreviations (e.g., "IS" in Table 1) appearing in the table should be written in full at the bottom note of the table.

Discussion

Line 371 – 389, pages 23 – 24: 1) "Limitations" discusses the shortcomings of the study itself, while "future directions" is not only to improve the shortcomings of the study but also to write about the aspects of in-depth research. Also, you should more clearly list the strengths of this study. Understanding the difference between the two, I believe you can write this section in a more logical way. I suggest dividing your opinions into “strengths and limitations” and “future directions” with subtitles and paragraphs. 2) The “conclusion” part of the study is missing.

This draft is a proper improvement, and I believe you have put a lot of effort into it. I hope you can make further revisions to increase the readability and standardization of the article.

Thank you and my best,

Your reviewer

Reviewer 2

All comments have been addressed.

Runtang Meng PhD

PLoS One, Guest Editor

Reviewers' comments:

Reviewer's Responses to Questions

**Comments to the Author**

1. If the authors have adequately addressed your comments raised in a previous round of review and you feel that this manuscript is now acceptable for publication, you may indicate that here to bypass the “Comments to the Author” section, enter your conflict of interest statement in the “Confidential to Editor” section, and submit your "Accept" recommendation.

Reviewer #1: (No Response)

Reviewer #2: All comments have been addressed

2. Is the manuscript technically sound, and do the data support the conclusions?

Reviewer #1: Partly

Reviewer #2: (No Response)

3. Has the statistical analysis been performed appropriately and rigorously? 

Reviewer #1: Yes

Reviewer #2: (No Response)

4. Have the authors made all data underlying the findings in their manuscript fully available?

Reviewer #1: Yes

Reviewer #2: (No Response)

5. Is the manuscript presented in an intelligible fashion and written in standard English?

Reviewer #1: Yes

Reviewer #2: (No Response)

6. Review Comments to the Author

Reviewer #1: ID: PONE-D-21-13329

Title: Do Alexithymia and Negative Affect Predict Poor Sleep Quality? The Moderating Role of Interoceptive Sensibility.

I appreciate your efforts to improve the manuscript and to respond to the comments made in the first review process. However, there are still minor issues that need to be addressed.

Comment: minor revision.

Detailed information:

Abstract

This section is still too long, especially the "Objectives" part, please simplify it appropriately.

Materials and methods

Participants

Line 167, page 7, “The university is in a large city”: This means nothing to me and the readers. I suggest removing it.

Line 173 – 174, page 8: All number digits should be unified. Check other places too.

This section is not very fluent and seems like several different parts spliced together. I would suggest combining and trimming some of the sentences to make them easier to read.

Measurements

Line 185-186, page 8; line 199, page 9; line 239, page 11: The reliability of this study should be placed in the "Results" section.

Procedure

Line 250-253, page 11: "NTD" or "NTDs"? Please express it uniformly.

Results

Line 267 – 269, page 12: Please state vital results in the tables and change these two sentences with "(Table 1)" and "(Table 2)", which may make the expression clearer.

Line 270 – 271, page 12: Please standardize the number of decimals appearing in the whole text.

Table 1 – Table 4: 1) All scientifical tables should be formatted as three-line tables. 2) All abbreviations (e.g., "IS" in Table 1) appearing in the table should be written in full at the bottom note of the table.

Discussion

Line 371 – 389, pages 23 – 24: 1) "Limitations" discusses the shortcomings of the study itself, while "future directions" is not only to improve the shortcomings of the study but also to write about the aspects of in-depth research. Also, you should more clearly list the strengths of this study. Understanding the difference between the two, I believe you can write this section in a more logical way. I suggest dividing your opinions into “strengths and limitations” and “future directions” with subtitles and paragraphs. 2) The “conclusion” part of the study is missing.

This draft is a proper improvement, and I believe you have put a lot of effort into it. I hope you can make further revisions to increase the readability and standardization of the article.

Thank you and my best,

Your reviewer

Reviewer #2: (No Response)

7. PLOS authors have the option to publish the peer review history of their article (what does this mean?). If published, this will include your full peer review and any attached files.

Reviewer #1: No

Reviewer #2: No

---

## [Author Response · Author response to Decision Letter 1]

14 Sep 2022

Responses to Reviewers

Reviewer 1 ID: PONE-D-21-13329 Title: Do Alexithymia and Negative Affect Predict Poor Sleep Quality? The Moderating Role of Interoceptive Sensibility. I appreciate your efforts to improve the manuscript and to respond to the comments made in the first review process. However, there are still minor issues that need to be addressed. 

Comment: minor revision. 

Detailed information: 

Abstract 

1. This section is still too long, especially the "Objectives" part, please simplify it appropriately. 

REPLY: We thank the reviewer for this suggestion. We have shortened the objective and conclusions. The abstract are 218 words after revision.

Line 27-55

Objectives

Emotion-related hyperarousal is an important core pathology of poor sleep. Studies investigating the interplay of alexithymia and affective experiences in determining sleep quality have yielded mixed results. To disentangle the inconsistency, this study examined the concurrent predictive power of alexithymia, and negative and positive affect, while incorporating interoceptive sensibility (IS) as a possible moderator.

Methods

A sample of 224 (70.1% were female) participants completed the Toronto Alexithymia Scale, Positive and Negative Affect Schedule, Pittsburgh Sleep Quality Index, Multidimensional Assessment of Interoceptive Awareness (MAIA), and Marlowe‐Crowne Social Desirability Scale (for controlling response bias) using paper and pencil. A two-stage cluster analysis of the MAIA was used to capture IS characteristics. Stepwise regression was conducted separately for each IS cluster.

Results

A three-group structure for IS characteristics was found. Higher alexithymia was predictive of poor sleep quality in the low IS group, while higher negative affect predicted poor sleep quality in the moderate and high IS groups. Additionally, alexithymia and positive affect were significantly different in the three IS groups, while negative affect and sleep quality were not.

Conclusions

Emotion and cognitive arousal may impact sleep quality differently in individuals with different levels of internal focusing ability, depending on physiological versus emotional self-conceptualization. The implications on pathological research, clinical intervention, study limitations and future directions are discussed.

Materials and methods Participants 

1. Line 167, page 7, “The university is in a large city”: This means nothing to me and the readers. I suggest removing it. 

REPLY: The authors thank reviewer for this suggestion. We have removed it.

2. Line 173 – 174, page 8: All number digits should be unified. Check other places too. This section is not very fluent and seems like several different parts spliced together. I would suggest combining and trimming some of the sentences to make them easier to read. 

REPLY: The authors thank reviewer for this suggestion. We have checked the number digits through the manuscript, and revised this section as following:

Line 160-171

The data are part of a larger project, the Human Project from Mind, Brain, and Learning (HPMBL) in Taiwan. HPMBL concerns comprehensive well-being of adults in urban life, and was approved by the Research Ethics Committee of Chengchi University (NCCU-REC-201810-I074). Online advertisements were posted on a university discussion webpage on social network media. The inclusion criteria for HPMBL were: 1) age 20 to 64; 2) native speakers of traditional Chinese; 3) normal vision (with or without correction); and 4) not diagnosed as having mental disorders that influence reality testing or cognitive ability, such as schizophrenia or dementia. A total of 249 participants participate the HPMBL; 25 participants had missing questionnaire data for this study. Hence, a final sample of 224 participants was included in this analysis. The mean age of final sample was 22.13 years (SD = 2.71), and there were 157 (70.10%) women and 67 (29.90%) men.

3. Measurements Line 185-186, page 8; line 199, page 9; line 239, page 11: The reliability of this study should be placed in the "Results" section. 

REPLY: The authors thank the reviewer for this suggestion. We have moved reliability of this study into the Results and renamed the first section as “Characteristics of measures and three IS clusters.”

Line 260-262

The overall Cronbach’s alpha of PSQI was 0.89; 0.74 for TAS; 0.84 for PA and 0.90 for NA; 0.91 for MAIA; 0.68 for Marlowe-Crowne Social Desirability Scale.

4. Procedure Line 250-253, page 11: "NTD" or "NTDs"? Please express it uniformly. 

REPLY: We are sorry for this error; it was unified as NTD after revision.

Results 

1. Line 267 – 269, page 12: Please state vital results in the tables and change these two sentences with "(Table 1)" and "(Table 2)", which may make the expression clearer. 

REPLY: The authors thank the reviewer for the suggestions. We have revised these two sentences. In addition, we noticed that in our previous revision, we added information according to three IS groups in Table 1, however mention our clustering in Table 3. Hence, we also restructured the Results and tables. Now, the Table 1 presents how the three group of IS constructed. The Table 2 shows the demographic and measure attributes of participants. The Table 3 presents correlations between all measures. 

Line 262-276

The overall Cronbach’s alpha of PSQI was 0.89; 0.74 for TAS; 0.84 for PANAS-PA and 0.90 for PANAS-NA; 0.91 for MAIA; 0.68 for Marlowe-Crowne Social Desirability Scale. According to the final centroids (Table 1), the three IS clusters were categorized as low, moderate, and high IS groups. There was no significant difference of demographic data between the three groups. TAS and PANAS-PA scores showed statistically significant difference, such that the high IS group exhibited lower TAS and higher PANAS-PA; PANAS-NA and PSQI were not significantly different among the groups (Table 2). We also tested the gender differences on all measures controlling for social desirability. Males are higher on the MAIA-Attention Regulation (F = 4.05, p = .045), MAIA-Self-Regulation (F = 4.39, p = .037), and TAS-EOT (F = 6.37, p = .012). All other measures were not significantly different from male to female.

After controlling for social desirability, except for the two reversed scales (N-D and N-W), the MAIA subscales were profoundly related to each other. TAS was associate with PANAS-PA and PANAS-NA; in contrast, PANAS-PA and PANAS-NA was not associated. PSQI was significantly associated with TAS, PANAS-PA, and PANAS-NA (Table 3).

2. Line 270 – 271, page 12: Please standardize the number of decimals appearing in the whole text. Table 1 – Table 4: 1) All scientifical tables should be formatted as three-line tables. 2) All abbreviations (e.g., "IS" in Table 1) appearing in the table should be written in full at the bottom note of the table. 

REPLY: The authors thank the reviewer for this remind. We have standardized the number of decimals as two (except for p values), and added “0” before decimal points when applicable. The formatting and footnote of tables were also adjusted. 

Discussion 

1. Line 371 – 389, pages 23 – 24: 1) "Limitations" discusses the shortcomings of the study itself, while "future directions" is not only to improve the shortcomings of the study but also to write about the aspects of in-depth research. Also, you should more clearly list the strengths of this study. Understanding the difference between the two, I believe you can write this section in a more logical way. I suggest dividing your opinions into “strengths and limitations” and “future directions” with subtitles and paragraphs. 

REPLY: The authors thank the reviewer and we have revised the discussion part per your important suggestion. The section “strengths and limitations” was revised, and a future direction section was added.

Line 369-401

There are several strengths and limitations of this study. By incorporating IS, this study raised fresh insights on psychopathology and interventions for sleep disturbances. The individual difference of IS provide a possibility to explain the inconsistent findings of alexithymia, negative affect and sleep problem. Accordingly, individualized intervention for sleep problems can be designed. Apart from the main focus of the study, we have also found a potential differentiation between PA and NA in their association strength with somatic feelings. The participants were community samples recruited through online advertisements. Hence, self-selection bias was unavoidable. A higher homogeneity of participants may also lead to other problems in addition to a lack of generalizability. We did not find significant gender differences in self-reported negative affectivity, alexithymia, or sleep quality, as in previous studies [71-73]. This may be due to the high homogeneity that overrides gender differences. In sum, this study’s ecological validity is not ideal. Although the regression results in our study were statistically significant, the R2s were all small, indicating a small effect of the studied variables. Sleep is a complex process influenced by the circadian, homeostatic, and arousal processes [2,74,75]. Emotion and alexithymia are only a part of the arousal process; hence, the contribution might in fact be small. 

Several directions can be applied in future studies. First, IS is a subcomponent of interoception, the “sensing the physiological condition of the body, as well as the representation of the internal state within the context of ongoing activities, and is closely associated with motivated action to homeostatically regulate the internal state” [76, p.693]. Other domains of interoception can be investigated. Similarly, there are multiple levels and measures of affect and sleep quality. Incorporating different levels of such construct, especially physiological assessments of interoception (e.g., brain imaging of insula, heart-evoked potential heartbeat detection task), affect (e.g., brain and autonomic reactions of emotion induction), and sleep (e.g., polysomnography) can provide important information. Second, emotion regulation is an important factor involved with arousal-related sleep problem intervention. Additionally, emotion regulation is embedded in the construct of alexithymia [13,14], intimately related to negative affective experience [32], and highly relevant to interoceptive awareness [36]. Future study may incorporate emotion regulation in addition to the constructs of this study. Third, sleep and interoception may interplay within aspects other than hyperarousal. For example, interoception is also directly linked to homeostasis [76], another determined process of sleep quality [77]. Some possibilities of the relations between sleep and interoception have been discussed, however more empirical studies are required to determine such interplay [78]. Beyond the relationship between sleep and interoception, our finding further suggests that individual differences of interoception is associated with the express of differential mechanism of sleep process. This phenomenon can be examined in homeostatic relevant aspects of sleep.

2. 2) The “conclusion” part of the study is missing. This draft is a proper improvement, and I believe you have put a lot of effort into it. I hope you can make further revisions to increase the readability and standardization of the article. Thank you and my best, Your reviewer 

REPLY: The authors thank the reviewer for the important suggestions and encouragement. A conclusion part was added per your suggestion. 

SLine 410-416

Conclusion

We tested a novel idea that how an individual process their own bodily sensations associates with sleep disturbances using self-reported questionnaires. Alexithymia predicts lower sleep quality among low IS group. In contrast, negative affectivity predicts lower sleep quality among moderate and high IS groups. Further examination of psychopathology and intervention of sleep disturbance incorporating IS should be implemented in future.

---

## [Editor Report · Decision Letter 2]

15 Sep 2022

Do Alexithymia and Negative Affect Predict Poor Sleep Quality? The Moderating Role of Interoceptive Sensibility

PONE-D-21-13329R2

Dear Dr. Yen,

We’re pleased to inform you that your manuscript has been judged scientifically suitable for publication and will be formally accepted for publication once it meets all outstanding technical requirements.

Kind regards,

Runtang Meng, Ph.D., M.M., M.B.B.S.

Guest Editor

PLOS ONE
---

## [Editor Report · Acceptance letter]

23 Sep 2022

PONE-D-21-13329R2 

Do Alexithymia and Negative Affect Predict Poor Sleep Quality? The Moderating Role of Interoceptive Sensibility 

Dear Dr. Yen:

I'm pleased to inform you that your manuscript has been deemed suitable for publication in PLOS ONE. Congratulations! Your manuscript is now with our production department. 

Kind regards, 

on behalf of

Dr. Runtang Meng 

Guest Editor

PLOS ONE